# Oxidative Stress Biomarkers among Schizophrenia Inpatients

**DOI:** 10.3390/brainsci13030490

**Published:** 2023-03-14

**Authors:** Magdalena Więdłocha, Natalia Zborowska, Piotr Marcinowicz, Weronika Dębowska, Marta Dębowska, Anna Zalewska, Mateusz Maciejczyk, Napoleon Waszkiewicz, Agata Szulc

**Affiliations:** 1Department of Psychiatry, Faculty of Health Sciences, Medical University of Warsaw, 02-091 Warsaw, Poland; nataliazborowska8@gmail.com (N.Z.);; 2Experimental Dentistry Laboratory, Medical University of Bialystok, 15-089 Bialystok, Poland; 3Department of Hygiene, Epidemiology and Ergonomics, Medical University of Bialystok, 15-089 Bialystok, Poland; 4Department of Psychiatry, Medical University of Bialystok, 16-070 Choroszcz, Poland

**Keywords:** oxidative stress, kynurenine pathway, schizophrenia

## Abstract

Background. Finding the associations between schizophrenia symptoms and the biomarkers of inflammation, oxidative stress and the kynurenine pathway may lead to the individualization of treatment and increase its effectiveness. Methods. The study group included 82 schizophrenia inpatients. The Positive and Negative Symptoms Scale (PANSS), the Brief Assessment of Cognition in Schizophrenia (BACS) and the Calgary Depression in Schizophrenia Scale were used for symptom evaluation. Biochemical analyses included oxidative stress parameters and brain-derived neurotrophic factor (BDNF). Results. Linear models revealed the following: (1) malondiadehyde (MDA), N-formylkynurenine (N-formKYN), advanced oxidation protein products (AOPP), advanced glycation end-products of proteins (AGE) and total oxidative status (TOS) levels are related to the PANSS-total score; (2) MDA, reduced glutathione (GSH) and BDNF levels are related to the PANSS-negative score; (3) TOS and kynurenine (KYN) levels are related to the PANSS-positive score; (4) levels of total antioxidant status (TAS) and AOPP along with the CDSS score are related to the BACS-total score; (5) TAS and N-formKYN levels are related to the BACS-working memory score. Conclusions. Oxidative stress biomarkers may be associated with the severity of schizophrenia symptoms in positive, negative and cognitive dimensions. The identification of biochemical markers associated with the specific symptom clusters may increase the understanding of biochemical profiles in schizophrenia patients.

## 1. Introduction

Despite advances in the knowledge of schizophrenia research, the efficacy of antipsychotic treatment is not satisfactory. It is estimated that approximately 30% of patients respond to treatment by achieving complete remission of symptoms, 30% manage to achieve partial remission while 20–30% do not respond to treatment [1]. Antipsychotics are much more effective against positive symptoms than against negative and cognitive symptoms. At the same time, it is the severity of the deficit symptoms and cognitive dysfunction that is the most important predictor of psychosocial functioning in schizophrenia [2].

The identification of sensitive and specific biomarkers that indicate susceptibility, which may help explain the clinical features, clinical response to treatment and improve therapy precision, remains one of the greatest challenges in schizophrenia research. Different profiles of gene expression abnormalities, epigenetic patterns, metabolic and inflammatory markers in the peripheral blood are found among patients with schizophrenia [3].

The involvement of inflammatory processes in the pathophysiology of schizophrenia is well documented. Many studies have focused on the search for disease biomarkers among proinflammatory and anti-inflammatory factors. It has been suggested that abnormalities of the immune system including cytokine profiles may be a biochemical endophenotype in some schizophrenia patient populations [4]. Abnormal cytokine levels are found in schizophrenic patients from the onset of the first episode psychosis (FEP), as well as in their first-degree relatives [5,6].

Pro-oxidant and antioxidant factors are also among the commonly assessed biomarkers in schizophrenia. Researchers report significant disease-related abnormalities suggesting increased oxidative stress (OS) and decreased activity of antioxidant mechanisms [7,8,9]. The increased production of reactive oxygen species (ROS) and reactive nitrogen species (RNS) as well as reduced antioxidant potential are considered risk factors for the development of schizophrenia. It was shown that different fractions of brain proteins exhibit signs of oxidative damage [10]. ROS stimulate the expression of genes encoding proinflammatory cytokines including tumor necrosis factor α (TNF-α), interleukin (IL) 1 and IL-6 and affect the activity of the nuclear factor (NF-kB) that determines proinflammatory activity [11]. There exist interrelationships between chronic OS and chronic inflammation that lead to impaired processes of apoptosis and neurogenesis and the occurrence of neurodegenerative changes which have a documented role in the etiopathogenesis of schizophrenia [12]. Craddock et al. found that in patients with schizophrenia, the intensity of the immune response was associated with the level of OS [13]. Moreover, antioxidant activity was shown to be decreased in schizophrenia patients [14,15,16].

Studies also indicated that metabolites of the kynurenine pathway (KP) may be a valuable biomarker of the disease process in schizophrenia [17,18]. The tryptophan metabolism activity within KP depends on the severity of inflammation [19]. KP metabolites directly or indirectly affect the oxidation-reduction status. This may be a pro-oxidant or antioxidant effect depending on the cell type, pH, exposure time and oxidation-reduction potential of the environment [20]. Under physiological conditions, the potentially adverse effects are balanced by oxidation-reduction balance mechanisms. The loss of this balance can lead to OS and cell destruction [17,18].

Another biomarker studied in patients with schizophrenia is brain-derived neurotrophic factor (BDNF). Its plasma and serum levels correlate with concentrations in the cerebrospinal fluid (CSF) and with concentrations of N-acetylaspartate, a marker of neuronal cortex integrity [21]. In schizophrenia, reduced BDNF levels have been found in the prefrontal cortex and hippocampus [22]. A study conducted by Belbasis et al. involving a large umbrella meta-analysis has shown that decreased BDNF levels are a significant factor associated with the increased risk of schizophrenia [23].

The influence of the above factors on specific dimensions of the schizophrenia features is being actively investigated, yet is not clearly defined [4,6,8,9]. The identification of biochemical markers associated with the specific symptom clusters may be the basis for the creation of a ‘biochemical profile’ to individualize treatment and increase its effectiveness. The assessment of peripheral biomarkers may also indicate the most appropriate adjuvant strategies to enhance the efficacy of schizophrenia treatment. The aim of this study was to determine the relationship between the levels of selected inflammatory, pro-oxidant and antioxidant markers; initial KP metabolites; BDNF; and symptom severity of the different dimensions of schizophrenia: positive, negative, cognitive and depressive.

## 2. Materials and Methods

### 2.1. Study Group

The study group consisted of 82 patients with a diagnosis of paranoid schizophrenia hospitalized in the general psychiatric wards of the Mazovian Specialist Health Centre in Pruszków between 2017 and 2019. Each patient gave written informed consent to participate in the study. All subjects went through at least 2 psychotic episodes. During the study, patients were under antipsychotic treatment in monotherapy.

Exclusion criteria included comorbid mental disorders, including organic disorders, intellectual disability, abuse of or addiction to alcohol and/or psychoactive substances (except nicotine and caffeine) or concomitant neurological, autoimmune, infectious or chronic somatic diseases. In addition, patients with present inflammatory markers in screening blood tests (C-reactive protein, CRP > 5mg/L and white blood count, WBC > 10,000/μL) were excluded from the study.

The study received a positive opinion from the Bioethics Committee at the Warsaw Medical University.

### 2.2. Sociodemographic and Clinical Data Collection

Data collected included age, gender, duration of illness (DI), number of psychotic episodes (NPE), body mass index (BMI), smoking and antipsychotic treatment used. To compare the doses of different antipsychotics (AP), the doses used were converted to chlorpromazine equivalents (CPZE) according to Woods et al. [24].

### 2.3. Clinical Scales Used to Assess the Severity of Psychopathological Symptoms

The Positive and Negative Symptoms Scale (PANSS) was used to assess the severity of positive (PANSS-P), negative (PANSS-N), general symptoms (PANSS-G) and the total score (PANSS-T) [25]. Cognitive functioning was assessed using the Brief Assessment of Cognition in Schizophrenia (BACS). It is a tool developed to assess cognitive functions in schizophrenia, validated with standardized batteries of cognitive function tests in patients with schizophrenia and in healthy individuals with respect to age, race and education. The BACS allows the examination of 6 domains that are particularly impaired in schizophrenia and have the greatest impact on functioning: verbal memory (BACS-VM), working memory (BACS-WM), motor speed (BACS-MS), verbal fluency (BACS-VF), attention and speed of information processing (BACS-ASP) and executive functions (BACS-EF) [25]. We also assessed the total BACS result (BACS-T). The results of each BACS subtest and BACS-T were standardized by calculating the deviation (Z-score) from the average result for healthy controls (0-score). The 0-score along with norms for age and gender were determined by the tool authors [26]. The Calgary Depression Scale for Schizophrenia (CDSS) was used to assess the severity of depressive symptoms in schizophrenia [27].

### 2.4. Blood Sample Collection and Storage

Venous blood samples were collected after overnight (12 h) fasting into an EDTA-containing tube. Subsequently, blood samples were centrifuged at 2000× *g* for 10 min at room temperature. The obtained serum was immediately frozen and stored at −80 °C until the biochemical analyses. All samples were analyzed in a single batch.

### 2.5. Biochemical Analyses

#### 2.5.1. IL-6

IL-6 levels were determined spectrophotometrically using ELISA. The diagnostic kit used was the *Human IL-6 Quantikine ELISA Kit according to the* manufacturer’s recommendations.

#### 2.5.2. Antioxidant Enzymes

The catalase (CAT) activity was determined colorimetrically in triplicate samples using a method based on measuring the rate of decomposition of hydrogen peroxide in phosphate buffer at pH 7.0 at 240 nm. One unit of CAT activity was defined as the amount of enzymes that degraded 1 mmol H_2_O_2_ in 1 min [28].

The glutathione peroxidase (GPx) activity was determined in triplicate samples using a colorimetric method according to Mansson-Rahemtull et al. involving the reduction of 5,5′-dithio-bis-2-nitrobenzoic acid to thionitrobenzoic acid, which was then reacted with OSCN- ions (hypothiocyanates): a product of oxidation of potassium thiocyanate by GPx. The decrease in absorbance was measured at 412 nm, and 5 absorbance measurements were taken every 30 s [29].

#### 2.5.3. Reduced Glutathione (GSH)

The GSH concentration was determined in duplicate samples using a colorimetric method based on the reduction of 5,5′-dithiobis-2-nitrobenzoic acid to 2-nitro-5-mercaptobenzoic acid under the influence of GSH contained in the test sample. The concentration of 2-nitro-5-mercaptobenzoic acid formed in the reaction was measured at 412 nm and calculated from a calibration curve determined for GSH solutions [30].

#### 2.5.4. Oxidation-Reduction Balance Parameters

The total antioxidant status (TAS) was determined in triplicates using a colorimetric method measuring changes in absorbance of an ABTS+ (3-ethylbenzothiazoline-6-sulphonic acid radical cation) solution under the influence of antioxidants contained in the test sample at 660 nm. The TAS was calculated from the standard curve for Trolox (6-hydroxy-2,5,7,8-tetramethylchroman-2-carboxylic acid) [31].

The total oxidative status (TOS) was determined in triplicate samples by a bichromatic method (560/800 nm) involving the oxidation of Fe ions^2+^ to Fe^3+^ in the presence of oxidants contained in the sample, followed by the detection of Fe^3+^ by xyleneol orange. The TOS concentration was presented as μmol H_2_O_2_ Equiv./L (micromolar hydrogen peroxide equivalent per liter) [32].

The oxidative stress index (OSI) was presented as the quotient of TOS to TAS and was expressed in %.

#### 2.5.5. Products of Oxidative Damage of Proteins and Lipids

The advanced oxidation protein products’ (AOPP) concentration was determined in duplicate samples by a colorimetric method measuring the oxidation capacity of iodide iodine at 340 nm. Serum samples were pre-diluted in phosphate-buffered saline at a ratio of 1:5 [33].

The advanced glycation end-products of proteins’ (AGE) concentration was determined by measuring the fluorescence characteristic of AGE derivatives (350 nm/440 nm) in duplicate samples. Serum samples were previously diluted in PBS solution in a ratio of 1:5 [33].

The malondialdehyde (MDA) concentration was determined in duplicate samples by the thiobarbituric acid colorimetric method. Absorbance was measured at 535 nm and 1,1′,3,3′-tetraethoxypropane was used as a standard [34].

#### 2.5.6. Protein Glyco-Oxidation Products

To assess dityrosine (DITYR), kynurenine (KYN) and N-formylkynurenine (N-formKYN) concentrations, samples were diluted in 0.1 M sulfuric acid at a volume ratio of 1:10. Fluorescence was measured at wavelengths 330/415 nm (DITYR), 365/480 nm (KYN), 325/434 nm (N-formKYN) and 95/340 (Trp) [35].

#### 2.5.7. Nitrosative Stress Parameters

The nitrogen oxide (NO) level was determined colorimetrically by measuring its stable metabolites NO_3_^−^ and NO_2_^−^ in a Griess reaction. Changes in optical density were measured at 543 nm [36].

The peroxynitrite (ONOO^−^) level was determined by a nitration reaction resulting in nitrophenols [37].

The 3-nitrotyrosine (3-NT) level was determined spectrophotometrically using ELISA. The Immundiagnostik AG diagnostic kit was used according to the manufacturer’s instructions.

#### 2.5.8. BDNF

BDNF levels were determined spectrophotometrically using an ELISA method. The *Total BDNF Quantikine ELISA Kit* was used according to the manufacturer’s recommendations.

### 2.6. Statistical Analysis

A statistical analysis was performed using the Statistica package version 13.3, scripts implemented in R, version 4.1.2 and Excel spreadsheets. Quantitative variables were described by the arithmetic mean with standard deviation, median and range and binary variables by percentages. The Shapiro–Wilk W-test was used to determine whether a quantitative variable came from a population with a normal distribution. In the univariate analysis, relationships between quantitative variables were determined by identifying Spearman correlations along with r-correlation coefficient values. In order to identify factors that were predictors of greater severity of psychopathological symptoms from the individual dimensions of schizophrenia, generalized linear models (GLM) were used. For all statistical calculations, the p-value of 0.05 was taken as the limit of significance. For biochemical markers, as these were not normally distributed, analyses have been performed using natural logarithm transformed values. As the study population size was relatively small, Type I and II errors had an increased risk of occurring, which has been accounted for where applicable.

## 3. Results

### 3.1. Characteristics of the Study Population

Demographic and clinical data are shown in Table 1. The study group included 82 participants: 57.3% males and 42.6% females. The median of illness duration and number of psychotic episodes were, respectively, 89 months and 4.5. The BMI mean and median values were in the upper limit of healthy weight range. Most of the study participants were tobacco smokers.

### 3.2. Results of Clinical Scales and Measurement of Biochemical Parameters in Study Population

Table 2 shows the means with SD, ranges, medians and 95%CI for the scores of the clinical scales and subscales in the study population. In the study population, all evaluated cognitive function scores were lower than the standards adopted for the healthy controls. Among all cognitive domains, working memory Z-scores were the highest. Table 3 shows the means with SD, ranges, medians and 95%CI for the values of each biochemical parameter in the study population.

### 3.3. Correlations between Assessed Parameters

The univariate analysis examined correlations between biochemical parameters and PANSS, BACS and CDSS scores and demographic data. An analysis of the results was conducted taking multiple comparisons into account, and the significance level value was adjusted using the Holm–Bonferroni method to account for Type 1 errors.

There was a high positive correlation of PANSS-P levels with TOS and OSI values and an average positive correlation with KYN levels (Table 4). There was also a high positive correlation of PANSS-N levels with MDA values and an average negative correlation with GSH levels (Table 5). We found a high positive correlation of PANSS-T levels with TOS and OSI values and an average positive correlation with IL-6 (Table 6). In the field of cognitive functions, there was an average positive correlation of BACS-WM scores with TAS values and an average negative correlation with N-formKYN levels (Table 7). Moreover, we found a high negative correlation of BACS-ASP and BACS-EF levels with ONOO- values and an average negative correlation with NO and AOPP levels (Table 8).

An analysis of correlations between BACS, PANSS and CDSS scores revealed only an average negative correlation of BACS-T with CDSS scores (Table 9). Among the correlations between PANSS, BACS and CDSS scores with demographic data, no significant correlations were observed.

### 3.4. Linear Model Describing the Relationship between Biochemical and Clinical Parameters and the Severity of Psychopathological Symptoms as Measured by the PANSS-T

The model was constructed to determine associations between the PANSS-T score and biochemical and clinical parameters. The model was built in a backward, stepwise manner optimizing the AIC value, and for the final model, an F-test was performed to assess model significance and R^2^ was calculated to indicate goodness of fit.

The final model consisted of five predictors: MDA, N-formKYN, AOPP, AGE and TOS. These had a significant (*p* < 0.05) effect on the PANSS-T score (Table 10), with the model being significant and of satisfactory fit. Based on the constructed model, higher values of MDA, N-formKYN, AOPP, AGE and TOS were associated with a higher PANSS-T score.

Additionally, DI, NPE, BMI, smoking, CPZE and AP were all included in the final model but had no significant impact on the PANSS-T score and did not influence other associations.

Subsequent linear models describing the relationships between biochemical and clinical parameters and symptom severity of each dimension of schizophrenia were constructed in the same manner as the model above.

### 3.5. Linear Model Describing the Relationships between Biochemical and Clinical Parameters and the Severity of Negative Symptoms as Measured by the PANSS-N Subscale

The model consisted of three predictors: MDA, BDNF and GSH. These had a significant (*p* < 0.05) effect on the PANSS-N score (Table 11). From the constructed model, higher values of MDA together with lower levels of GSH and BDNF were associated with a higher PANSS-N score.

Additionally, DI, NPE, BMI, smoking, CPZE and AP were all included in the final model but had no significant impact on the PANSS-T score and did not influence other associations.

### 3.6. Linear Model Describing the Relationship between Biochemical and Clinical Parameters and the Severity of Positive Symptoms as Measured by the PANSS-P Subscale

The model consisted of two predictors, TOS and KYN. These had a significant (*p* < 0.05) effect on the PANSS-P score (Table 12). From the constructed model, higher TOS and KYN values were associated with higher PANSS-P scores.

Additionally, DI, NPE, BMI, smoking, CPZE and AP were all included in the final model but had no significant impact on the PANSS-T score and did not influence other associations.

### 3.7. Linear Model Describing the Relationship between Biochemical and Clinical Parameters and the BACS-WM Result

The model consisted of two predictors, TAS and N-formKYN. These had a significant (*p* < 0.05) effect on the BACS-WM score (Table 13). From the constructed model, higher TAS and lower N-formKYN values were associated with a higher BACS-WM score.

Additionally, DI, NPE, BMI, smoking, CPZE and AP were all included in the final model but had no significant impact on the PANSS-T score and did not influence other associations.

### 3.8. Linear Model Describing the Relationship between Biochemical and Clinical Parameters and the BACS-T

The linear model consisted of three predictors: TAS, AOPP and CDSS. These had a significant (*p* < 0.05) effect on the BACS-T score (Table 14). From the constructed model, higher TAS values and lower AOPP and CDSS values were associated with a higher BACS-T score.

Additionally, DI, NPE, BMI, smoking, CPZE and AP were all included in the final model but had no significant impact on the PANSS-T score and did not influence other associations.

Statistical models constructed using the GLM to describe the relationship between biochemical and clinical parameters and PANSS-G subscale scores, BACS-ASP, BACS-EF, BACS-VF, BACS-VM, BACS-MS and CDSS were characterized by a coefficient of determination R^2^< 0.3. This indicates an unsatisfactory fit of the model to data; therefore, these models were not included in the results.

## 4. Discussion

### 4.1. Inflammation and Oxidative-Nitrosative Stress and the Severity of Psychopathological Symptoms in Schizophrenia

This study showed that a higher severity of OS expressed by OSI and higher concentrations of products of protein (AOPP, AGE, N-formKYN) and lipid (MDA) oxidation processes are associated with greater severity of psychopathological symptoms in schizophrenia.

The OSI parameter identifies a shift in the oxidation-reduction balance towards oxidative processes. As reported by Juchnowicz et al. when compared with HC, schizophrenia patients, regardless of stage, exhibited several times higher TOS and OSI values, and these parameters were considered a risk marker for the development of the disease. However, in contrast to the results of the present study, no association was found between OSI levels and the severity of schizophrenia symptoms [38]. Other researchers found higher OSI levels in patients with marked deficit symptoms, in remission and with chronic illness compared to patients without deficit symptoms and those who did not achieve remission [39,40]. In the study by Sertan Copoglu et al. an association between oxidative DNA damage and the severity of schizophrenia symptoms was demonstrated. Patients not in remission showed higher levels of TOS, OSI and 8-hydroxydeoxyguanine (8-OHdG), a marker of oxidative DNA damage, and reduced levels of TAS. In contrast, patients in remission showed a positive correlation between TOS and OSI levels and 8-OH-dG [40].

AOPP is a sensitive marker of oxidative damage to proteins, particularly albumin, as well as fibrinogen and lipoproteins [41]. Oxidized proteins have an ability to activate the inflammatory response and can also induce a sudden release of large amounts of ROS by neutrophils and the production of chemotactic factors for inflammatory cells. They also stimulate the production of IL-8 and TNF-α by monocytes. Due to the above properties, AOPP is considered a marker and mediator of the proinflammatory effect of OS [42]. In relation to the results of the present study, it can therefore be speculated that inflammation may be a factor associated with the severity of psychopathological symptoms. This was also suggested by other reports. Zhang et al. stated that the cytokine system’s dysregulation and oxidative stress may induce clinical symptoms of schizophrenia [43]. Liemburg et al. found that CRP was associated with positive and negative symptom severity in a large sample of outpatients with chronic schizophrenia. Higher concentrations of the proinflammatory cytokines such as TNF-α and IL-6 were related to a deficit syndrome, while the TNF-α level was associated with negative symptom severity [44]. Guidara et al. also demonstrated an effect of AOPP levels on the severity of schizophrenia symptoms [45]. On the other hand, in a study by Juchnowicz et al. this parameter was related to the age of the patients and the duration of the illness, but no relationship was found with symptom severity [46].

AGEs, including methylglyoxal and 3-deoxyglucosone, are formed during the oxidation of lipids, glucose and amino acids. They are highly reactive and, as with AOPPs, tend to accumulate, generate ROS and induce inflammation [45]. The binding of AGEs to the membrane receptor of macrophages, myocytes, neurons and other cells results in the increased synthesis of proinflammatory cytokines and secondary production of ROS. AGEs also decrease the antioxidant potential by modifying and inactivating CAT, GPx and superoxide dismutase (SOD) [47]. Juchnowicz et al. found higher levels of AGE in patients with schizophrenia compared to the healthy controls [46]. A systematic review by Kouidrat et al. also found an accumulation of AGEs in patients with schizophrenia [48].

The level of MDA, a product of lipid peroxidation, along with the concentration of protein oxidation and glycation products, was associated in our study with greater severity of pathological symptoms in the study group of schizophrenia patients. The results from studies to date show an increase in lipid peroxidation in schizophrenia, and this is also supported by meta-analyses [8,49,50]. Arvindagshan et al. also showed that the severity of lipid peroxidation, as measured by the level of end-products of the process, was associated with symptom severity in schizophrenic patients [51]. Recent research indicated a positive correlation between the PANSS-P score and MDA as well as CRP levels in FEP drug-naïve patients. Dudzinska et al. suggested that MDA may be an early indicator of ongoing low-grade inflammation [52]. Lipid peroxidation causes structural and functional damage to cell membrane phospholipids and polysaturated fatty acids [53]. Guidara et al. suggested that the specific behavioral symptomatology of schizophrenia may be related to arachidonic acid oxidative damage and its consequences for central nervous system (CNS) neurochemistry [45].

The present study also found an association between the severity of psychopathological symptoms in PANSS and levels of N-formKYN and KYN: compounds that are products of tryptophan oxidation. Juchnowicz et al. also found an association between greater severity of psychopathological symptoms in PANSS and higher KYN and lower TAS levels in patients with schizophrenia [38]. Metabolism within KP provides neuroactive kynurenine derivatives that may significantly influence the pathophysiology of schizophrenia by modulating dopaminergic, glutamatergic and nicotinergic transmission and disrupting the oxidative-reduction balance. Kynurenic acid (KYNA), an N-methyl-D-aspartate receptor (NMDAR) antagonist, has a neuroprotective effect at normal concentrations, whereas at elevated concentrations it leads to excessive NMDAR blockade, contributing to psychotic symptoms and cognitive deficits [18]. Other metabolites of KP, such as 3-hydroxykynurenine and the NMDAR agonist quinolinic acid, have neurotoxic and neurodegenerative effects [19]. In our study, the levels of only the initial KP metabolites were assessed, so it is not possible to draw conclusions regarding a direct effect of all KP metabolites on the clinical picture in schizophrenia. However, it appears that the association between N-formKYN and KYN levels and the severity of psychopathological symptoms may be explained by increased tryptophan metabolism within KP and possibly increased levels of downstream, neurotoxic metabolites.

### 4.2. Biochemical Markers Associated with Increased Positive Symptoms in Schizophrenia

A multivariate regression model analysis showed that higher OS as measured by TOS together with higher KYN concentrations were a predictor of higher positive symptom severity in the study group. As described above, KP metabolites may have pro-oxidant and neurotoxic effects. Previous studies indicate that there is a relationship between inflammatory markers, OS and KP metabolites and the severity of positive symptoms.

It was shown that the severity of positive symptoms in schizophrenia is associated with higher levels of KYN and ferric reducing ability of plasma (FRAP) [38]. KYN was also postulated to be a biomarker in monitoring the progress of treatment [54]. SOD was also found to be negatively correlated with positive symptom severity, but this was not confirmed in the meta-analysis by Flatow et al. [7,55,56]. Dietrich-Muszalska et al. instead showed an association between the severity of positive symptoms and the severity of lipid peroxidation [57]. They also found a correlation between levels of the proinflammatory cytokines IL-1, IL-7 and IL-8 and the severity of delusions [58].

### 4.3. Biochemical Markers Associated with the Severity of Negative Symptoms in Schizophrenia

Statistical modeling of the obtained data showed that a shift in the oxidation-reduction balance towards oxidative processes expressed by increased lipid peroxidation along with lower GSH and BDNF levels is associated with greater severity of negative symptoms in schizophrenia. Previous studies confirmed the association between antioxidant potential and the severity of negative symptoms in schizophrenia. Li et al. showed a negative correlation between TAS levels and negative symptoms in patients with first episode psychosis (FEP). Moreover, they found that the presence of OS at the onset of psychosis influenced the subsequent course of the illness, especially the development of negative symptoms [59]. The results of the study by Albayrak et al. confirmed these relationships. Patients with persistent negative symptoms were found to have lower levels of TAS and higher levels of OSI compared to patients with non-deficit schizophrenia and healthy controls (HC). They also showed that higher CAT levels in patients with schizophrenia were associated with a lower risk of negative symptoms, shorter duration of illness and fewer episodes [39]. The results of a study by Juchnowicz et al. also showed an association of FRAP, CAT and dityrosine levels with the severity of negative symptoms [38].

The association of negative symptoms with GSH levels shown in this study is consistent with the results of previous studies and the meta-analysis by Flatow et al. [7,9]. In the study by Matsuzawa et al. greater negative symptom severity was associated with lower GSH levels in the posterior medial frontal cortex of patients with schizophrenia [60]. Maes et al. also found a greater reduction in GSH levels in schizophrenia patients with predominantly negative symptoms [61]. Decreased levels of GSH and the efficiency of the antioxidant system along with increased sensitivity to OS may influence various pathophysiological processes found in schizophrenia, including the impairment of dopaminergic neurotransmission and NMDAR responses to glutamate [62]. GSH has a direct effect on glutamatergic neurotransmission through interaction with NMDARs, and NMDAR activity enhances and regulates GSH metabolism [63]. GSH levels increase in response to the glutamate-dependent excitatory activity of parvalbumin-induced GABAergic interneurons (PVIs) in the prefrontal cortex, leading to its downregulation and protecting neurons from OS. Decreased NMDAR activity contributes to GSH deficits and increased OS in the CNS, and in turn, even transient GSH deficiency leads to decreased NMDAR activity [64]. As a consequence, the inhibitory activity of PVIs as well as their number is reduced, which result in an excitatory–inhibitory imbalance in the CNS [63,65]. GSH deficiency also contributes to myelination abnormalities, which may have a significant impact on the deterioration of cognitive function in schizophrenia [3]. A negative correlation was also found between the activity of GPx, an enzyme belonging to the glutathione system, and the severity of cerebral atrophy in patients with chronic schizophrenia [66]. Impaired antioxidant activity and its influence on structural and functional changes in the CNS may explain the association between lower GSH levels and greater severity of negative symptoms.

The study also found a negative effect of MDA and a positive effect of BDNF on the severity of deficit symptoms in schizophrenia. Elevated levels of MDA and a decreased proportion of polyunsaturated fatty acids were reported in patients with negative symptoms, suggesting that oxidative damage may implicate this clinical dimension [67]. In addition, the association of reduced BDNF levels with the severity of negative symptoms in schizophrenia is supported by some studies [68,69]. Others do not show this relationship [22,70]. In patients with bipolar affective disorder, a negative correlation was found between plasma BDNF and lipid peroxidation product levels, which may indicate that BDNF protects neurons from damage resulting from OS [71]. The neuroprotective role of BDNF is documented in many studies. The antiapoptotic effect is associated with the activation of the intracellular signaling cascade by the receptor of the tyrosine kinase B family (TrkB), towards which BDNF has high affinity [72]. In an in vitro study, BDNF was found to protect cortical neurons from NMDA- and H_2_O_2_-induced apoptosis by inhibiting the mitogen-activated kinase (MAPK) cascade [73]. The use of exogenous BDNF significantly inhibited the loss of dopaminergic neurons in the black matter caused by oxidative damage to cells [74].

### 4.4. Biochemical Markers Associated with the Severity of Cognitive Impairment in Schizophrenia

We assessed six domains of cognitive function: verbal memory, working memory, verbal fluency, motor speed, attention and speed of information processing and executive functions. The results of our study suggest that higher levels of cognitive impairment in schizophrenia may be associated with more intense protein oxidation as measured by AOPP levels, lower plasma antioxidant potential as measured by TAS and greater severity of depressive symptoms. The predictive value of TAS as a biomarker of cognitive impairment in schizophrenia has been shown. Martinez-Cengotitabengoa et al. found a positive correlation of the TAS level with total cognitive performance both at FEP and after 2 years of illness [75].

In the study population, the levels of all assessed cognitive functions were below the norms adopted for healthy individuals. The largest deficit was in WM, the dysfunction of which is considered one of the primary disorders of the schizophrenic process [76]. A significant effect of lower TAS and higher N-formKYN concentrations on greater severity of WM impairment was found. Martinez-Cengotitabengoa et al. also showed a correlation between TAS levels and working memory performance in a group of patients with non-affective psychosis both at FEP and after a follow-up of 2 years [75]. The shift of tryptophan metabolism towards KP associated with the production of neurotoxic metabolites may have a significant impact on WM impairment in schizophrenia. Many studies have documented the adverse effects of KP metabolites on cognitive function in schizophrenia. Kindler et al. showed that higher values of the KYN/tryptophan ratio were associated with reduced volumes of the dorsolateral prefrontal cortex, a region crucial for normal WM function, and more severe attention disorders and increased levels of proinflammatory cytokines [77]. Koola et al. showed that peripheral levels of KYN and KYNA can be an indicator of the degree of cognitive deterioration and a useful marker for monitoring treatment effects [54].

EF as well as ASP also show significant deterioration in schizophrenia [70,76,78]. These manifest as difficulties with stimulus selection and a tendency to process irrelevant information, which can lead to misinterpretation of percepts. Some authors indicated a positive correlation of ASP and EF performance with BDNF levels [79,80]. The severity of cognitive dysfunction in schizophrenia, including attention and EF, has also been found to be associated with immune activation and cytokine levels. Perkins et al. documented the correlation of proinflammatory IL-1, IL-7 and IL-8 levels with the severity of attention deficits [58]. In contrast, analyses using statistical models showed an association of EF disorders with interactions between BDNF and IL-8, BDNF and TNF-α, BDNF and SOD and BDNF and MDA [70,81]. This study did not find any associations between ASP or EF impairment severity and IL-6 levels, duration of illness, BDNF levels or other symptom severity. In our results, the levels of NO, ONOO^−^ and AOPP correlated negatively with ASP and EF scores, but statistical modeling did not include these parameters as predictors of EF or ASP scores. Nevertheless, the associations we found may imply a negative effect of OS and NS on these cognitive domains. As AOPP is considered a marker and a mediator of the proinflammatory effect of OS, it may confirm an important role of inflammation in ASP and EF impairment in schizophrenia [45]. The deleterious effects of NO are mainly related to the highly reactive products of its metabolism, including ONOO^−^, which reacts with downstream molecules, leading to increased levels of RNS, ROS and increased lipid and protein oxidation. The negative effect of NO on assessed cognitive functions may be a consequence of increased glutamatergic excitotoxicity. NO is a mediator of NMDAR activation, and its concentration reflects glutamatergic transmission in the CNS [82]. Wang et al. found a negative correlation of NO levels with information processing speed, working memory performance and verbal learning [83]. The results of this study, like ours, indicated a negative effect of NO and related NS on cognitive function in schizophrenia.

### 4.5. Biochemical Markers Associated with Severity of Depressive Symptoms in Schizophrenia

The study found that higher BDNF levels were associated with lower depressive symptoms in schizophrenia. BDNF’s role in the pathophysiology of depression is well known, and lower serum levels of the factor were found in people with major depressive disorders compared to HCs [84]. Some researchers pointed to a beneficial effect of BDNF on the severity of depressive symptoms in schizophrenia, which they link to the neuroprotective effect of this factor [85]. Interestingly, there are studies showing a negative correlation between BDNF levels and the severity of depressive symptoms in schizophrenia [86,87]. This is explained as the result of a compensatory increase in BDNF synthesis in response to oxidative stress and increased proinflammatory cytokine activity [86].

### 4.6. Limitations

When interpreting the results presented in this study, it is important to take into account the limitations. The study did not include an HC group. Therefore, it cannot be concluded whether the measured parameters’ levels significantly differed between schizophrenia patients and HC.

The second limitation is a relatively small size of the study group. Moreover, it was comprised of hospitalized patients only, which may imply a greater severity of symptoms and poorer functioning than in the general population of schizophrenic patients. All study subjects were on antipsychotic treatment, taking different antipsychotics and the duration of treatment was variable. Previous studies are inconsistent regarding the effects of antipsychotics on oxidative stress markers. Some researchers postulated that typical antipsychotics increase oxidants and decrease antioxidants, while atypical antipsychotics cause the opposite effect [88,89]. On the other hand, it was shown that atypical antipsychotics may increase OS and decrease antioxidant activity [90,91]. However, studies of previously untreated individuals with FEP show that abnormalities in the biochemical factors studied appear early in the course of schizophrenia, which suggests that they may be a part of disease pathophysiology independently from pharmacotherapy [59]. Our study showed no significant differences in the measured parameters between the groups of patients taking different antipsychotics. However, an influence of pharmacotherapy cannot be excluded.

The majority of the study group were smokers. Although no significant differences in study factors were found between the smoking and non-smoking patients, a significant effect of nicotinism on the parameters of inflammation and oxidation-reduction balance cannot be excluded. Studies have shown an association between chronic smoking and levels of some OS and NS parameters, including AGE and NO [92,93]. The physiological process of aging also involves changes in factors related to the oxidation-reduction balance. With age, there is a decrease in the activity of antioxidant enzymes and an increase in the oxidative potential and the concentration of lipid peroxidation products [53]. It has also been shown that age-related decreases in GSH levels are more strongly expressed in schizophrenia. The study did not show any correlation between the age of the patients and the parameters assessed. However, the influence of aging-related processes on the results and the correlations found also cannot be excluded. The study assessed only biochemical parameters from peripheral blood. Based on the available data, it was assumed that the concentration/activity of the examined factors in peripheral blood correlated with values in the CNS [94,95]. In order to confirm the correlations shown and the validity of the conclusions drawn from them, it would be worth extending the methodology to neuroimaging studies.

Moreover, the measured biochemical parameters are not specific to schizophrenia and may be involved in the pathophysiology of various psychiatric and somatic diseases. To minimize the influence of other medical conditions, subjects with comorbid psychiatric disorders, somatic diseases and clinical or laboratory signs of inflammation were not included in the study. In order to increase the sensitivity and specificity of the correlations found, in addition to the analysis of associations between individual factors, statistical modeling methods were applied to the data. The levels of studied biochemical parameters may be influenced by factors that were not included in the study, such as diet or physical activity [46].

## 5. Conclusions

1. Increased oxidative stress, protein oxidation and glycation processes and lipid peroxidation are associated with greater severity of psychopathological symptoms in schizophrenia.

2. Higher levels of oxidative stress together with higher levels of kynurenine are related to greater severity of positive symptoms in schizophrenia. More intense lipid peroxidation together with lower levels of GSH and BDNF correspond with more expressed negative symptoms.

3. Greater cognitive impairment severity in schizophrenia is associated with lower antioxidant potential along with increased protein oxidation and more severe depressive symptoms. Greater working memory impairment in schizophrenia is linked with lower antioxidant potential and higher N-formylkynurenine levels.

## Figures and Tables

**Table 1 brainsci-13-00490-t001:** Demographic and clinical data in the study population (N = 82). (DI–duration of illness, NPE–number of psychotic episodes, BMI–body mass index, CPZE–chlorpromazine equivalents, AP–antipsychotic treatment).

		Mean (SD)	Min–Max	Median	95%CI
Age		36.57 (8.278)	22–54	37.5	[7.176;9.782]
DI [months]		84.42 (41.483)	4–166	89	[35.961;49.024]
NPE		4.75 (1.674)	2–9	4.5	[1.451;212.4]
BMI		24.27 (2.402)	20.2–28.6	24.45	[2.083;1.978]
CPZE		422.7 (97.02)	239.5–600.6	402.3	[84.11;114.7]
Gender	Female	35 (42.6%)			
	Male	47 (57.3%)			
AP	Haloperidol	7 (8.5%)			
	Clozapine	20 (24.3%)			
	Olanzapine	33 (40.2%)			
	Risperidone	22 (26.8%)			
Smoking	Yes	56 (68.2%)			
	No	26 (31.8%)			

**Table 2 brainsci-13-00490-t002:** Results of clinical scales and subscales for the study population (N = 82).

	Mean (SD)	Min–Max	Median	95%CI
PANSS-G	40.88 (6.023)	28–57	41	[5.222;7.118]
PANSS-P	25.34 (6.181)	16–36	25	[5.358;7.304]
PANSS-N	20 (3.428)	13–28	20	[2.972;4.051]
PANSS-T	86.18 (8.592)	66–107	85.5	[7.449;10.15]
BACS-WM	−1.8369 (1.111)	−3.7304–0.0356	−1.8293	[0.963;1.313]
BACS-ASP	−0.6279 (0.5714)	−2.25–0.47	−0.5579	[0.4953;0.6752]
BACS-EF	−1.1355 (0.4024)	−2.48–0.3	−1.1261	[0.3488;0.4755]
BACS-VF	−0.6687 (0.8223)	−1.99–0.96	−0.6575	[0.7129;0.9718]
BACS-VM	−0.5829 (0.7106)	−1.86–0.92	−0.6423	[0.616;0.8398]
BACS-MS	−0.192 (0.6722)	−1.35–1.12	−0.1574	[0.5827;0.7944]
BACS-T	−1.2183 (0.5124)	−2.2822–0.1427	−1.2479	[0.4442;0.6055]
CDSS	12.13 (2.297)	6–7	12	[1.992;2.715]

**Table 3 brainsci-13-00490-t003:** Results of the analysis of individual biochemical parameters in the study population (N = 82).

	Mean (SD)	Min–Max	Median	95%CI
IL-6	150.7 (152)	25.99−557.2	71.74	[131.7;179.6]
CAT	150.7 (152)	25.99−557.2	71.74	[131.7;179.6]
GPx	18.79 (7.104)	10.16−35.86	17.13	[6.159;8.396]
GSH	0.4514	(0.0643)	2863–0.6563	[0.0557;0.076]
TAS	1.652 (0.1159)	1.348−1.971	1.637	[0.1005;0.137]
TOS	11.24 (5.486)	2.052−22.09	11.16	[4.756;6.483]
OSI	6.854 (3.391)	1.254−13.26	6.661	[2.94;4.008]
AOPP	3764.1 (1649.8)	1032.1−9788.1	3439.1	[1430.2;1949.7]
AGE	55.51 (11.64)	19.35−98.37	55	[10.09;13.76]
MDA	0.3696 (0.2393)	0.132−0.9871	0.2473	[0.2075;0.2828]
DITYR	56.2 (11.83)	19.97−97.42	55.84	[10.26;13.98]
KYN	89.07 (14.86)	50.08−129.9	91.87	[12.88;17.56]
N-formKYN	1177 (478.4)	187−2057.7	1313.7	[414.7;565.4]
Try	476.6 (108.2)	133.6−979.2	486.4	[93.81;127.9]
NO	60.13 (32.67)	21.03−229.6	51.54	[28.32;38.61]
ONOO^−^	12.93 (4.772)	7.101−35.15	12.02	[4.136;5.639]
3-NT	1493.4 (1249.4)	10.62−7428	1168.6	[1083.1;1476.5]
BDNF	0.6448 (0.6453)	0.3098−3.189	0.3509	[0.5594;0.7625]

**Table 4 brainsci-13-00490-t004:** Statistically significant correlations between PANSS-P and biochemical parameters.

	TOS	OSI	KYN
PANSS-P			
coefficient	0.663	0.652	0.443
*p*	<0.001	<0.001	<0.001
95%CI	[0.521;0.769]	[0.507;0.761]	[0.250;0.602]

**Table 5 brainsci-13-00490-t005:** Statistically significant correlations between PANSS-N and biochemical parameters.

	GSH	MDA
PANSS-N		
coefficient	−0.413	0.519
*p*	<0.001	<0.001
95%CI	[−0.578;−0.216]	[0.340;0.661]

**Table 6 brainsci-13-00490-t006:** Statistically significant correlations between PANSS-T and biochemical parameters.

	IL-6	TOS	OSI
PANSS-T			
coefficient	0.344	0.530	0.531
*p*	<0.002	<0.001	<0.001
95%CI	[0.137;0.522]	[0.354;0.670]	[0.355;0.671]

**Table 7 brainsci-13-00490-t007:** Statistically significant correlations between BACS-WM and biochemical parameters.

	TAS	N-formKYN
BACS-WM		
coefficient	0.483	−0.398
*p*	<0.001	<0.001
95%CI	[0.297;0.634]	[−0.566;−0.198]

**Table 8 brainsci-13-00490-t008:** Statistically significant correlations between BACS-ASP, BACS-EF and biochemical parameters.

	AOPP	NO	ONOO-
BACS-ASP			
coefficient	−0.678	−0.445	−0.493
*p*	<0.001	<0.001	<0.001
95%CI	[−0.780;−0.540]	[−0.604;−0.253]	[−0.642;−0.310]
BACS-EF			
coefficient	−0.727	−0.397	−0.434
*p*	<0.001	<0.001	<0.001
95%CI	[−0.815;−0.606]	[−0.565;−0.197]	[−0.595;−0.240]

**Table 9 brainsci-13-00490-t009:** Statistically significant correlations between BACS-T and CDSS.

	CDSS
BACS-T	
coefficient	−0.049
*p*	<0.001
95%CI	[−0.640;−0.307]

**Table 10 brainsci-13-00490-t010:** Final linear model of PANSS-t score (AIC 320.378; F = 19.023, *p* < 0.001; R^2^ = 0.550).

Factor	Estimation	Standard Error	95%CI	*p*
Free expression	55.33705	4.053109	47.26458;63.40951	<0.0001
MDA	10.82218	2.880301	0.00091;0.00251	<0.0003
N-formKYN	0.00438	0.001417	5.08556;16.55879	<0.0027
AOPP	0.00171	0.000404	0.01279;0.24595	<0.0001
AGE	0.12937	0.058534	0.47058;0.96391	<0.0300
TOS	0.71724	0.123847	0.00156;0.00721	<0.0001

**Table 11 brainsci-13-00490-t011:** Final linear model of PANSS-N score (AIC 212.842; F = 15.374, *p* < 0.001; R^2^ = 0.571).

Factor	Estimation	Standard Error	95%CI	*p*
Free expression	21.57604	1.285538	19.01673;24.13535	0.0001
BDNF	−1.01343	0.478248	−1.96554;−0.06131	0.0372
MDA	5.83179	1.340924	3.16221;8.50136	0.0002
GSH	−1.22412	0.390273	−2.00109;−0.44715	0.0024

**Table 12 brainsci-13-00490-t012:** Final linear model of PANSS-P score (AIC 209.882; F = 42.350, *p* < 0.001; R^2^= 0.517).

Factor	Estimation	Standard Error	95%CI	*p*
Free expression	7.088859	2.975499	1.16627;13.01144	0.0196
TOS	0.691285	0.089440	0.05200;0.18344	0.0001
KYN	0.117720	0.033016	0.51326;0.86931	0.0006

**Table 13 brainsci-13-00490-t013:** Final linear model of BACS-WM outcome (AIC 54.171, F = 15.685, *p* < 0.001; R^2^ = 0.373).

Factor	Estimation	Standard Error	95%CI	*p*
Free expression	−6.52634	1.665466	−10.74561;−4.21494	0.0001
TAS	3.66830	0.923046	1.37451;5.72593	0.0001
N-formKYN	−0.00059	0.000223	−0.00107;−0.00017	0.0104

**Table 14 brainsci-13-00490-t014:** Final linear model of BACS-T score (AIC 173.203, F = 29.194, *p* < 0.001; R2 = 0.529).

Factor	Estimation	Standard Error	95%CI	*p*
Free expression	−0.844449	0.329856	−2.09839;0.409497	0.0183
AOPP	−0.000110	0.000025	−0.15514;−0.08485	0.0004
CDSS	−0.119996	0.017651	−0.00016;−0.00006	0.0002
TAS	0.906267	0.346496	0.21644;1.59608	0.0106

## Data Availability

Not applicable.

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
