# Peer review of "Oxidative Stress Biomarkers among Schizophrenia Inpatients"

_brainsci, 2023, doi:10.3390/brainsci13030490_

Round 1

Reviewer 1 Report

This is a cross sectional study assessing associations between a range of biomarkers and psychopathology in 82 hospitalized patients with schizophrenia.

I have some major concerns about the framing current knowledge about schizophrenia and how your findings add to our knowledge so far.

Line 32 Despite the increasing understanding of the etiopathogenesis of schizophrenia and the implementation of drugs with new, more specific mechanisms of action, the efficacy of antipsychotic treatment is not satisfactory.

This description of schizophrenia is exaggerated in my perspective. There are no replicated findings in biological research increasing the etiopathogenesis of schizophrenia. Drugs are no more specific now than chlorpromazine was 70 years ago, but still modify symptoms broadly and not the underlying disease.

Line 87 The aim of our study was to determine the relationship between levels of selected inflammatory macrometers, pro-oxidant and antioxidant exponents, initial KP metabolites, BDNF and symptom severity of the different dimensions of schizophrenia: positive, negative, cognitive and depressive.

In the conclusion, you abstain from the word association, and use “potential predictors”. I suggest you use the same terminology, and abstain from speculating about predictors, due to the cross sectional design of this study.

I also have concerns about the risk of spurious findings due to lack of pre-registration and lack of adjustment for multiple testing. You have allowed yourselves several possible ways to calculate the outcomes, and therefore increase the risk of P-hacking. Also, small sample size increases the risk of type 1 and type 2 errors, which should be mentioned.

In the tables, I would appreciate confidence intervals and p-values for all reported associations.

Finally, I am wondering why you did not adjust for confounding. Several domains, including comorbidity, medication and lifestyle is known to affect oxidative stress!

Author Response

Thank you for your review and all valuable comments. The point-by-point response to the comments is provided in the attached file. 

Reviewer 2 Report

In this study, the authors analysed many oxidative stress related markers in patients with schizophrenia and linked these markers with severity of symptoms.

Although the authors covered many aspects of oxidative stress, from ROS consequences on lipids and proteins, to antioxidant capacities, this study lack novelty as compared to the already available literature in the field. Also, the need of analysing so many markers are debatable, as no real mechanistic hypotheses are made between the markers, and the amount of statistical analyses increase the risk for false positive.

Moreover, many information are lacking to fully understand and appreciate the conclusions made by the authors:

-       The characterization of patients: patients are hospitalized, but it is not clear whether they are at the early psychosis phase, first episod or chronic. The duration of illness is indicated in the demographic table, but it is not clear whether it is days or months. The age of this population is rather high, with a mean at 36, so it seems to be rather chronic patients. This point is critical in order to understand the aim of the study: if they are chronic patients, the interest of finding  a link between peripheral biomarkers and severity of symptoms is not so obvious to me. If patients are already affected, no need of peripheral biomarkers.

-       The samples in which the biomarkers are analysed: is it whole blood, blood cells, plasma, serum? Depending on the type of samples, the interpretation of the results are different.

-       The blood processing protocol: no information at all. While looking at oxidative stress markers, the blood processing procedure, and the storage are critical steps.

-       Missing information on the statistical analyses: it is not mentioned whether the authors performed correction for the multiple comparison. Indeed, when performing so many correlation, the risk of false postive is very high. Also, did the authors include confounding factors, such as age, BMI, duration of illness, gender and smoking status? All these factors can influence the correlations. Especially, the BMI is rather high, which is often the case in psychiatric patients, but have a great influence on oxidative stress markers and inflammation.

-       Page 6: what is PANSS-C?

-       Table 13: why did the authors put CDSS score in the linear regression with biological markers to explain the BACS-T score? As both cognitive measured are linked, of course CDSS will be associated with BACS-T.

-       It is a pity that these analyses were not conducted in healthy controls, to compare whether increased oxidative stress markers are observed in the patients. 

All these comments should be implement in the manuscript for a better understanding of the reader.

The discussion is rather long, and makes a lot of hypotheses and assumptions, which are a bit too far from the conclusion that we can make from this study. I would recommend shorten especially the first paragraph (p9-11) to concentrate on the real conclusion of their data. Also, many references from pioneer in the field are missing.

Author Response

(The authors gave the same response as above.)
